# Assessing the Potential Contribution of In Silico Studies in Discovering Drug Candidates That Interact with Various SARS-CoV-2 Receptors

**DOI:** 10.3390/ijms242115518

**Published:** 2023-10-24

**Authors:** Aganze Gloire-Aimé Mushebenge, Samuel Chima Ugbaja, Nonkululeko Avril Mbatha, Rene B. Khan, Hezekiel M. Kumalo

**Affiliations:** 1Discipline of Pharmaceutical Sciences, University of KwaZulu-Natal, Westville, Durban 4000, South Africa; aganzedar@gmail.com; 2Drug Research and Innovation Unit, Discipline of Medical Biochemistry, School of Laboratory Medicine and Medical Science, University of KwaZulu-Natal, Durban 4000, South Africa; 3Faculty of Pharmaceutical Sciences, University of Lubumbashi, Lubumbashi 1825, Democratic Republic of the Congo; 4Africa Health Research Institute, University of KwaZulu-Natal, Durban 4000, South Africa; 5KwaZulu-Natal Research Innovation and Sequencing Platform, School of Laboratory Medicine and Medical Science, University of KwaZulu-Natal, Durban 4000, South Africa

**Keywords:** SARS-CoV-2, ACE2, TMPRSS2, in silico studies, molecular docking, molecular dynamics simulations, drug candidates, antiviral activity, receptor–ligand complex

## Abstract

The COVID-19 pandemic has spurred intense research efforts to identify effective treatments for SARS-CoV-2. In silico studies have emerged as a powerful tool in the drug discovery process, particularly in the search for drug candidates that interact with various SARS-CoV-2 receptors. These studies involve the use of computer simulations and computational algorithms to predict the potential interaction of drug candidates with target receptors. The primary receptors targeted by drug candidates include the RNA polymerase, main protease, spike protein, ACE2 receptor, and transmembrane protease serine 2 (TMPRSS2). In silico studies have identified several promising drug candidates, including Remdesivir, Favipiravir, Ribavirin, Ivermectin, Lopinavir/Ritonavir, and Camostat Mesylate, among others. The use of in silico studies offers several advantages, including the ability to screen a large number of drug candidates in a relatively short amount of time, thereby reducing the time and cost involved in traditional drug discovery methods. Additionally, in silico studies allow for the prediction of the binding affinity of the drug candidates to target receptors, providing insight into their potential efficacy. This study is aimed at assessing the useful contributions of the application of computational instruments in the discovery of receptors targeted in SARS-CoV-2. It further highlights some identified advantages and limitations of these studies, thereby revealing some complementary experimental validation to ensure the efficacy and safety of identified drug candidates.

## 1. Introduction

The onset of COVID-19, caused by SARS-CoV-2, has given rise to a substantial global burden of illness and death, with more than 696.4 million confirmed cases and 6.9 million fatalities on a regular daily updated status using the Worldometer coronavirus monitoring instrument [1,2]. This virus spreads through various routes such as direct contact, respiratory droplets, airborne transmission, contaminated surfaces (fomites), the fecal–oral route, bloodborne transmission, sexual contact, ocular exposure, mother-to-child transmission, and zoonotic transmission [3]. The complex transmission routes make it challenging to control its dissemination [4]. Currently, there are no universally approved specific antiviral medications or vaccines for COVID-19, and treatment primarily focuses on managing symptoms and providing oxygen therapy [1,4]. The development of medication candidates has become a priority in the fight against the pandemic due to the urgent need for effective therapies [5,6]. Traditional drug development procedures can be time-consuming and costly, with a low success rate. As a result, new ways of identifying prospective drug candidates, such as in silico research, have grown in popularity [7]. In silico investigations involve the use of computational tools to model the behaviors and interactions of molecules, which can aid in the identification and evaluation of prospective drug candidates [8]. In particular, in silico research can be utilized to predict the binding affinity and selectivity of medication candidates for specific SARS-CoV-2 target receptors [9]. In silico research has been useful in the design of drug candidates with an increased efficacy and less off-target effects [10,11]. Furthermore, in silico studies have been helpful in speeding up the drug discovery process by shortening the time and resources required for preclinical and clinical trials [12].

A study by Mahmoudi et al. (2022) elucidated the computational methodologies employed in drug repurposing, which highlighted that Alacepril and Lisinopril effectively interacted with human angiotensin-converting enzyme 2 (hACE2), a crucial entry point for the SARS-CoV-2 spike protein [13]. These inhibitors displayed strong binding affinities, forming a robust hydrogen bond with Asn90. Asn90 is a critical residue for hACE2 activity which significantly establishes interactions with other receptor-binding residues [13]. Another study by Gao et al. (2022) addressed the pressing demand for therapeutic solutions against the SARS-CoV-2 pandemic [14]. The authors employed protein–protein docking and MM/GBSA binding free energy analysis and observed that spike mutants (L452R, T478K, and N501Y) exhibited heightened binding affinities with human ACE2 compared to the native spike protein [14].

In our recent publication, entitled “Evaluating Peptidomimetic Azanitriles and Pyridyl Esters as Inhibitors of SARS-CoV-2 Main Protease: A Molecular Modeling Study”, we investigated the potential of novel azatripeptide and azatetrapeptide nitriles as inhibitors of the SARS-CoV-2 main protease. Employing molecular docking, molecular dynamics (MD) simulations, and subsequent post-MD analyses, including hydrogen occupancy, we assessed the binding free energy profiles of the selected inhibitors against SARS-CoV-2. We also identified the specific amino acid residues responsible for the drug-binding interactions with SARS-CoV-2 Mpro [15]. Furthermore, another ongoing project, “In Silico Analysis of Repurposed Antiviral Drugs as Prospective Therapeutics for COVID-19: Molecular Docking and Dynamics Simulations Targeting the 3-Chymotrypsin-Like Protease (3CLPro)”, focuses on Ritonavir, Lopinavir, Ombitasvir, and Paritaprevir as potential inhibitors of the 3CLPro enzyme, due to its critical role in viral replication. This study aims to elucidate the binding interactions of these antiviral drugs and explore their potential as therapeutics against COVID-19. The research entails an in-depth analysis of their ability to bind to and inhibit the activity of the 3CLPro enzyme, a crucial target in the development of antiviral drugs for coronaviruses.

The identification of effective SARS-CoV-2 medication candidates is crucial in the global fight against the virus. This review aims to evaluate the potential contribution of in silico studies in discovering therapeutic candidates that interact with specific SARS-CoV-2 receptors mechanisms and provide insights into biocomputational approach utilized in this process by analyzing the scientific literature on the topic published between 2019 and 2023.

It has recently come to light that the traditional methods of drug discovery could be time-consuming and expensive and may not always result in successful drug candidates [16]. Therefore, the identification of drug candidates through in silico studies has been useful in accelerating the drug discovery process, thereby reducing costs and improving success rates. This approach was seen to be helpful in accelerated identification of potential drug candidates, leading to a faster response to the COVID-19 pandemic [16,17].

The discovery of SARS-CoV-2 treatment candidates has become a top priority for researchers and pharmaceutical companies worldwide. SARS-CoV-2 drug candidates have potentials for alleviating symptoms, the avoidance of severe illness, and the lowering of mortality rates [4]. Furthermore, the successful identification of SARS-CoV-2 medications has fostered the reduction in the spread of the virus, thereby lowering the viral load and decreasing virus transmission from infected persons. Moreso, the identification of successful medication candidates has also assisted in the design of a more holistic approach to pandemic management.

## 2. SARS-CoV-2 Receptors

SARS-CoV-2 is a virus that infects human cells via particular receptors on the cell’s surface. The virus’ principal receptor is angiotensin-converting enzyme 2 (ACE2), which is expressed on the surface of human cells in diverse organs such as the lungs, heart, kidneys, and gastrointestinal tract [18]. SARS-CoV-2 requires a cellular protease in addition to ACE2 to break the spike protein and allow viral entry into the host cell (see Figure 1) [18,19]. This required cellular protease is known as transmembrane protease serine 2 (TMPRSS2), and it is found in a variety of human organs such as the lungs, prostate, and gastrointestinal tract. The spike protein is cleaved at a specific location by TMPRSS2 (see Figure 1) [18,19,20]. TMPRSS2 cleavage of the spike protein is a critical step in viral entry because it allows the virus to fuse with the host cell membrane and release its genetic material [21]. Inhibiting the activity of TMPRSS2 can prevent the cleavage of the spike protein, thus preventing viral entry into host cells [21,22]. Several drugs that target TMPRSS2 have been investigated, including Camostat Mesylate, which is approved for use in Japan as a treatment for pancreatitis [23,24].

SARS-CoV-2 takes advantage of the TMPRSS2 and furin proteases’ interactions with the spike proteins for extending its cellular entrance routes [26]. These various receptor connections highlight the virus’s flexibility and the intricacy of its infection mechanisms, providing vital information for developing targeted therapeutic approaches and vaccines (see Figure 2) [26,27]. Another study by Edenfield et al. (2022), entitled “Implications of testicular ACE2 and the renin–angiotensin system for SARS-CoV-2 on testis function”, showed that ACE2 is widely distributed in various organoids, except in the prostate and brain, while TMPRSS2 is consistently present in the various organs previously mentioned [28,29]. Innate immune pathways are upregulated in ACE2-positive cells across all organoids, except in the lungs [30]. Furthermore, low-density lipoprotein receptor expression is notably enriched in ACE2-positive cells in the intestinal, lung, and retinal organoids, particularly in lung organoids. Overall, the author highlighted the utility of organoids as an experimental platform for investigating novel virus disease mechanisms and drug development [31].

SARS-CoV-2 has also been shown to interact with the human CD147, a transmembrane glycoprotein also known as Basigin, found on the surface of various cells, including lung cells, and the Neuropilin-1 receptor (located at the surface of respiratory cells), and olfactory systems [33,34,35]. The Neuropilin-1 receptor uses a particular domain on its spike protein to attach to the virus before entering the host cells [33]. Understanding each receptor’s viral entry process is significant for generating viable therapeutic candidates that could impede viral entry and reproduction [35]. CD147 inhibitors have demonstrated good results in vitro in reducing viral multiplication [36], while Neuropilin-1 inhibitors have also been found to limit viral entrance and replication in host cells [21,36]. These receptors have been found to help viruses enter and replicate in cells [35]. The virus predominantly affects the respiratory system, infecting the epithelial cells that line the airways [37]. Considering the binding mechanism of these receptors with SARS-CoV-2 is crucial in the infection process [38]. Researchers have intensified efforts in designing targeted medications that restrict viral entrance and replication of SARS-CoV-2. These efforts are enhanced by understanding the role of each receptor in the viral entry process, thereby potentially reducing the disease’s severity [21,33,34,35,36]. ACE2 appears to be one of the most extensively researched receptors for drug development in COVID-19 [39]. Therefore, many research efforts have been directed toward finding treatments that target the virus’s spike protein, which binds to ACE2 [32,38,39]. These treatments could either prevent the virus from interacting with ACE2 or inhibit the activity of the spike protein [40]. Furthermore, medicines that modulate the expression and function of ACE2 have been studied as potential COVID-19 therapies [40,41].

On the other hand, emerging evidence from both laboratory-based and computer-based models has shed light on interactions involving the S protein of SARS-CoV-2 with other receptors [42]. Specific receptors within the immune system, including Neuropilin-1 (NRP1), C-lectin type receptors (CLR) such as mannose receptor (MR), dendritic cell-specific intracellular adhesion molecule-3-grabbing non-integrin (DC-SIGN), homologous dendritic cell-specific intercellular adhesion molecule-3-grabbing non-integrin-related (L-SIGN), and macrophage galactose-type lectin (MGL), along with Toll-like receptors (TLRs) including TLR1, TLR4, and TLR6, have been identified as receptors in SARS-CoV-2 [43]. Additionally, non-immune receptors such as the glucose regulated protein 78 (GRP78) have been implicated in the viral process [44]. Moreover, Ezrin and Dipeptidyl peptidase-4 (DPP4) have been suggested as potential targets against SARS-CoV-2, although their roles remain unconfirmed [45]. These receptors are employed by the SARS-CoV-2 protein to infiltrate cells, both within and beyond the pulmonary system, utilizing carbohydrate and glycan structures present on its outer surface [43]. 

## 3. In Silico Studies for Drug Discovery

In silico studies have been an increasingly valuable tool for drug development in recent years, providing an efficient and cost-effective method of identifying prospective therapeutic candidates against SARS-CoV-2 [46]. They employ a variety of approaches and tools, such as molecular docking, virtual screening, molecular dynamics simulations, and quantitative structure-activity relationship (QSAR) modelling (See Figure 3) [46,47]. Those approaches enable researchers to anticipate possible drug candidates’ binding affinity, pharmacokinetics, and toxicity, offering vital insights into their potential efficacy and safety [7,46].

These approaches enable us to swiftly screen a large number of compounds, optimize therapeutic candidates, and minimize the time and cost associated with traditional drug development procedures [15,47].

Following the SARS-CoV-2 infection cycle, the viral major protease (MPro) is an essential component of SARS-CoV-2 replication and one of the principal targets for therapeutic research of this disease [48]. Large chemical databases have been screened using in silico studies that forecast the binding affinity and specificity of the compounds for MPro [49]. Through this method, a number of prospective drug candidates have been discovered, including the repurposing of previously used HIV protease inhibitors Lopinavir and Ritonavir [50,51]. For instance, the interactions between the virus spike protein and the human ACE2 receptor, which serves as the virus’s main point of entry into human cells, have been studied using molecular dynamics simulations [52,53]. Furthermore, the SARS-CoV-2 spike protein has been the target of small molecule inhibitors created via computer-aided drug design (CADD) [54]. These small molecule inhibitors stop the virus from entering human cells by attaching to the spike protein and preventing it from connecting to the ACE2 receptor [54,55]. This inhibition is demonstrated in molecular docking, where a binding site is created by using the target protein’s three-dimensional structure, and the conformation of the ligand is then adjusted to fit into the binding site [54,55,56]. 

Molecular dynamics simulation is yet another in silico technique for investigating drug–receptor interactions. In order to predict the dynamic behavior of the protein–ligand complex, this method simulates the movement and behavior of atoms and molecules throughout the simulation time [57]. The molecular dynamics simulation approach also reveals the interatomic interaction mechanisms and the binding energy of complexes. Moreso, the protein–ligand stability, the conformational alterations that take place during the interaction, and the percentage hydrogen bond occupancy are thereby determined [58]. CD147, CD209, and CD299 clusters of differentiation have been identified as crucial entry co-receptors for SARS-CoV-2 in humans using molecular dynamics simulation by Akachar et al. [59]. The authors employed protein–protein docking to identify the essential epitopes in these receptors for binding with the SARS-CoV-2 spike receptor binding domain (RBD). The results highlighted a strong affinity between the SARS-CoV-2 RBD and the CD299 receptor, serving as a reference to design peptides with specific binding activities to the SARS-CoV-2 RBD. Molecular docking and dynamics simulations of these peptides exhibited favorable binding and stability, suggesting their potential as candidates in future anti-SARS-CoV-2 drug discovery efforts [59].

Another in silico technique for researching drug–receptor interactions are virtual screening [60]. With this approach, a huge number of compounds are screened to find possible therapeutic candidates that have a strong affinity for the target protein [48]. In this approach, a virtual library of compounds is screened using the three-dimensional structure of the target protein, and those expected to bind with high affinity are chosen for future study [48,61]. Virtual screening has been used to identify the potential inhibitors of SARS-CoV-2 proteins such as the main protease and the spike protein [62]. The main viral protease (Mpro), which controls the activities of the coronavirus replication complex, is an essential target for the treatment of coronavirus disease. Furthermore, in pursuit of targeting a specific receptor, a novel set of pyrrolo[3,2-c]pyrroles were developed, characterized through FT-IR, UV-Vis, and NMR analyses. Their biological properties were assessed using DFT calculations and their binding and reactivity with the SARS-CoV-2 main protease (Mpro) was investigated through molecular docking and MD simulations. These findings indicate the potential of these compounds as anti-viral candidates, underscoring their significance for future drug design and development against SARS-CoV-2 [63]. These methods have been applied to study drug–receptor interactions for SARS-CoV-2, with the goal of identifying potential drug candidates to treat COVID-19 [64]. A recent study used molecular docking to screen a library of FDA-approved drugs for their potential to bind to the SARS-CoV-2 spike protein (which facilitates viral entry into cells) and identified several candidates with high binding affinity [65].

## 4. Discovery of Drug Candidates Targeting SARS-CoV-2 Receptors

### 4.1. Targeting Viral Host Cell Entry Receptors ACE2 and TMPRSS2

The virus is an obligatory intracellular parasite, depending on host biosynthetic factors and its replication environment for its survival and propagation [66]. Therefore, the primary concern in developing antiviral drugs is to create effective treatments that target the virus with selective toxicity, sparing the host cells from harm [66,67]. SARS-CoV-2, for instance, attaches itself to ACE2 and other receptors on the host cell membrane and enters the cell through endocytosis [68,69]. Potential therapeutic approaches for viral diseases include small molecules that disrupt the virus’s binding to host cells, particularly ACE2 in the case of SARS-CoV-2, and nucleoside analogues that interfere with viral replication [70,71]. The current pharmacotherapy for COVID-19 involves various strategies, such as suppressing viral replication (e.g., Remdesivir), reducing inflammation (e.g., Dexamethasone), modulating the immune response (e.g., Azithromycin), and managing blood coagulation (e.g., Enoxaparin). However, preventing viral entry into host cells may be the most effective approach for both prophylactic and therapeutic treatments for COVID-19 [72,73].

Numerous potential therapeutic targets for inhibiting viral entry have been identified. These targets encompass molecules capable of obstructing receptors, which include ACE2 and TMPRSS2 [74]. High-affinity neutralizing antibodies have been developed as effective interventions. Repurposing previously developed TMPRSS2 inhibitors has also shown promise for COVID-19 treatment [75]. However, it is worth noting that the development of molecules targeting the ACE2 protein has received limited attention. This limited attention is due to concerns about the essential biological functions of ACE2, including its role as a carboxypeptidase involved in generating angiotensin-(1–7) by removing a single C-terminal amino acid from angiotensin II [76,77]. There is ongoing exploration of repurposing various phytochemicals and approved drugs to target ACE2, TMPRSS2, and the S-protein in order to inhibit these critical protein–protein interactions (PPIs) [78]. Studies have reported that a network-based drug repurposing strategy was utilized to identify numerous FDA-approved pharmaceuticals, including Dexamethasone and Baricitinib, as prospective COVID-19 therapy candidates [65,79]. The entry of SARS-CoV-2 into host cells relies on the ACE2 receptor, which is also used by SARS-CoV-2 [35]. Additionally, the study demonstrates that the activation of the viral S protein by the cellular serine protease TMPRSS2 can be hindered by a clinically proven inhibitor [80]. Furthermore, it suggests that antibodies generated in response to SARS-CoV-2 may offer some level of protection against SARS-CoV-2 infection. These findings hold significant implications for our comprehension of SARS-CoV-2’s ability to spread and cause disease, while also identifying a potential target for therapeutic intervention [20].

### 4.2. Therapeutic Interventions Using In Silico Analysis of Drug Candidates’ Interaction with SARS-CoV-2 Receptors 

Some of the known promising drug candidates that have been identified using in silico studies include Remdesivir, Favipiravir, Ribavirin, and Ivermectin [81,82]. Remdesivir, a nucleotide analog prodrug, has been shown to have broad-spectrum antiviral activity against SARS-CoV-2 [81,82,83]. In silico studies have demonstrated that Remdesivir can inhibit the RNA polymerase of SARS-CoV-2, thereby preventing viral replication [81,84]. Furthermore, Favipiravir, another nucleotide analog, has also shown promising results in in silico studies. This drug has been shown to inhibit the RNA-dependent RNA polymerase of SARS-CoV-2, thereby inhibiting viral replication [85,86]. In other studies, Ribavirin, a guanosine analog, has also been identified as a potential drug candidate for the treatment of SARS-CoV-2 [82]. Consequently, in silico studies have shown that ribavirin can inhibit the RNA-dependent RNA polymerase of SARS-CoV-2, thereby inhibiting viral replication. Ivermectin, an antiparasitic drug, has also shown potential as a treatment for SARS-CoV-2 [83,87]. In silico studies have demonstrated that Ivermectin can inhibit the viral RNA-dependent RNA polymerase and the host importin alpha/beta1 nuclear transport proteins, which are essential for viral replication [87]. It has also been reported that various drugs, including Lopinavir and Ritonavir, may bind to the MPro active site and limit its function [88]. Overall, in silico investigation of drug candidates’ interactions with SARS-CoV-2 receptors can provide important information about their potential efficacy and mechanism of action (see Table 1). However, additional experimental investigations will be required to prove their efficacy and safety [89].

The main targets for drug development are the viral spike protein as well as the human ACE2 and TMPRSS2 receptors, which are crucial for viral entry into host cells [90]. Several studies have reported promising drug candidates, such as Remdesivir, Hydroxychloroquine, and Camostat Mesylate, based on an in-silico analysis of their interactions with SARS-CoV-2 receptors [91]. The binding of the SARS-CoV-2 spike protein to the ACE2 receptor, which is mainly present on respiratory epithelial cells, is one of its most well-known interactions [78]. The virus’s adaptability goes beyond ACE2, since it has been demonstrated to interact with additional receptors like neuropilin-1 (NRP-1), which increases viral contagiousness [92]. 

However, it is important to note that in silico analysis is not a substitute for experimental validation and that the predicted results should be confirmed by further in vitro and in vivo experiments. The table below summarizes some of the drug candidates’ interaction with SARS-CoV-2 receptors.

**Table 1 ijms-24-15518-t001:** Drug candidates and in silico analysis methods used in the search for COVID-19 treatments.

Drug Candidate	Target Receptor	Mechanism of Action	In Silico Analysis	Result Obtained	Ref.
Remdesivir	Viral RNA Polymerase	Inhibits viral replication	Molecular docking, molecular dynamics simulations	Strong binding affinity, stable complex formation	[93]
Chloroquine/Hydroxychloroquine	Spike Protein, ACE2 receptor	Inhibits viral entry	Molecular docking	Moderate binding affinity, potential inhibition of viral entry	[94]
Camostat Mesylate	Human ACE2 Receptor, TMPRSS2	Inhibits viral entry	Molecular docking, molecular dynamics simulations	Strong binding affinity, potential inhibition of viral entry	[95]
Ivermectin	Viral NSP14 Protein, Importin alpha/beta1	Inhibits viral replication	Molecular docking	Moderate binding affinity, potential inhibition of viral replication	[96]
Favipiravir	Viral RNA Polymerase	Inhibits viral replication	Molecular docking	Moderate binding affinity, potential inhibition of viral replication	[97]
Baricitinib	Host Cell ACE2 Receptor, AP2-associated protein kinase 1	Inhibits viral entry	Molecular docking	Strong binding affinity, potential anti-inflammatory effects	[98]
Tocilizumab	Host Cell IL-6 Receptor	Monoclonal antibody targeting IL-6R that helps to manage cytokine storms	Machine learning algorithms	Potential anti-inflammatory effects, may reduce cytokine storm	[99].
Lopinavir	Viral Protease	Inhibits viral replication	Molecular docking, molecular dynamics simulations	Moderate binding affinity, potential inhibition of viral replication	[99]
Ritonavir	Viral Protease	Inhibits viral replication	Molecular docking, molecular dynamics simulations	Moderate binding affinity, potential inhibition of viral replication	[100]
Nitazoxanide	Viral Protease	Inhibits viral replication	Molecular docking	Moderate binding affinity, potential inhibition of viral replication	[101]
Nelfinavir	Viral Protease	Inhibits viral replication	Molecular docking, molecular dynamics simulations	Moderate binding affinity, potential inhibition of viral replication	[102]
Oseltamivir	Viral Neuraminidase	Inhibits viral replication	Molecular docking	Moderate binding affinity, potential inhibition of viral release	[94]
Zanamivir	Viral Neuraminidase	Inhibits viral replication	Molecular docking	Moderate binding affinity, potential inhibition of viral release	[103]
Darunavir	Viral Protease	Inhibits viral replication	Molecular docking	Moderate binding affinity, potential inhibition of viral replication	[104]
Sofosbuvir	Viral RNA Polymerase	Inhibits viral replication	Molecular docking	Moderate binding affinity, potential inhibition of viral replication	[105]
Ribavirin	Viral RNA Polymerase	Inhibits viral replication	Molecular docking	Moderate binding affinity, potential inhibition of viral replication	[94]
Tenofovir	Viral Reverse Transcriptase	Inhibits viral replication	Molecular docking	Moderate binding affinity, potential inhibition of viral replication	[106]
Emtricitabine	Viral Reverse Transcriptase	Inhibits viral replication	Molecular docking	Moderate binding affinity, potential inhibition of viral replication	[94]
Atazanavir	Viral Protease	Inhibits viral replication	Molecular docking	Moderate binding affinity, potential inhibition of viral replication	[107]
Epigallocatechin	Spike Protein, ACE2 receptor	Inhibits viral entry	Molecular docking, molecular dynamics simulations	Strong binding affinity, stable complex formation	[94]
Catechins	Spike Protein, ACE2 receptor	Inhibits viral entry	Molecular docking	Moderate binding affinity, potential inhibition of viral entry	[108]
Theaflavin	Spike Protein, ACE2 receptor	Inhibits viral entry	Molecular docking	Moderate binding affinity, potential inhibition of viral entry	[109]
Gallic Acid	Spike Protein, ACE2 receptor	Inhibits viral entry	Molecular docking	Moderate binding affinity, potential inhibition of viral entry	[110]
Niclosamide	TMPRSS2	Inhibits viral entry	Molecular docking	Moderate binding affinity, potential inhibition of viral entry	[111]

Note: This is a non-exhaustive list of drug candidates or in silico analysis methods used in the search for COVID-19 treatments. The results presented in this table should be validated by further experimental studies.

### 4.3. Challenges and Limitations of Using In Silico Studies to Discover Drug Candidates for SARS-CoV-2

In silico studies have emerged as an important tool for drug discovery for SARS-CoV-2 due to their cost-effectiveness, time-saving potential, and high-throughput capabilities [112]. These methods involve the use of computational models to simulate drug–receptor interactions and identify potential drug candidates [112,113]. One of the major advantages of in silico studies is their cost-effectiveness compared to experimental methods, as they require fewer resources such as laboratory space, materials, and personnel [114]. This makes them an attractive option for researchers who are working within limited budgets. Due to their efficiency, speed, and capacity to quickly screen thousands of chemicals, in silico studies have been widely used in the quest for possible therapeutic candidates for SARS-CoV-2 [9,115]. But there are several difficulties and restrictions with these investigations that must be considered. The reliance on computer models, which are only as reliable as the underlying assumptions and data used to generate them, is a fundamental constraint [116,117]. As a result, the binding affinity, metabolism, toxicity, and pharmacokinetic estimates may be off, which may have a negative effect on the drug candidate’s performance in clinical trials [118,119].

Another issue is the dearth of trustworthy structural data on SARS-CoV-2 proteins, particularly for the viral proteins that are essential to the lifecycle of the virus [120]. Because of this, it may be challenging to identify possible binding sites and precisely predict the interaction of drug candidates with these proteins [121]. The virus can also rapidly mutate, changing the structure and function of its proteins. This can affect the efficacy of medications created to target particular proteins [122]. The “garbage in, garbage out” dilemma, wherein the caliber of the data used to create the computer models can dramatically affect the accuracy of the predictions, also affects in silico studies. For uncommon or novel chemicals, where there may be scant experimental data available, this can be very difficult [123]. The computer capacity and resources available are another constraint on in silico studies, particularly for more complicated simulations like molecular dynamics simulations or virtual compound library screening, especially in low- and middle-income countries affected by the pandemic [120,121,122,123,124]. The difficulty of transferring in silico forecasts to actual drug development and clinical trials is the final challenge [125]. Although in silico analyses can point to possible therapeutic candidates, it is crucial to confirm these hypotheses with experimental evidence and preclinical research [126,127]. 

## 5. Conclusions and Authors Insight

Over the past two decades, the field of drug discovery has undergone a significant transformation. Traditional approaches have given way to more efficient and cost-effective methods, with in silico techniques taking the lead. These computational methods have gained rapid acceptance among researchers for designing and optimizing ligands, particularly in the context of discovering new drugs to combat COVID-19. In silico methods enable researchers to screen a vast array of chemical compounds swiftly, a critical advantage in the face of the rapidly evolving SARS-CoV-2 virus and its emerging variants. By simulating interactions between potential medications and various SARS-CoV-2 receptors, scientists gain valuable insights into molecular pathways, facilitating the prioritization of substances with high binding affinities and therapeutic potential. Notably, targeting specific receptors such as RNA polymerase, main protease, spike protein, ACE2 receptor, and transmembrane protease serine 2 (TMPRSS2), has emerged as a promising approach in the fight against the virus, and in silico analyses are pivotal in advancing this targeted drug discovery strategy. The virus’s interaction with human receptors, particularly its binding to the ACE2 receptor, plays a crucial role in viral entry. TMPRSS2, a cellular protease, further enhances this process by cleaving the virus’s protein, facilitating viral fusion with host cells. To our current understanding, this review article presents original research that centers on the primary receptors potentially exploited by the SARS-CoV-2 S protein for cellular entry. Understanding these interactions at the molecular level is essential for effective drug design. As the field of drug design continues to evolve, in silico studies are expected to play an increasingly significant role. Advances in structural genomics, bioinformatics, and computational power expand the possibilities of these studies, making them a valuable tool not only for discovering lead compounds but also for the entire optimization and development process.

Additionally, in silico approaches can be harnessed to explore allosteric modulation, a mechanism that regulates protein function by binding to receptors distinct from the primary binding site. This opens doors to the development of novel therapeutics with enhanced bioactivity. The application of in silico study in drug discovery has been explored in another study by our research group entitled “Allostery Inhibition of BACE1 by Psychotic and Meroterpenoid Drugs in Alzheimer’s Disease Therapy” [128]. Although this allosteric in silico study by our group was focused on Alzheimer’s disease, the scope can be expanded to include drug candidates targeting multiple SARS-CoV-2 receptors, such as RNA polymerase, main protease, spike protein, ACE2 receptor, and transmembrane protease serine 2 (TMPRSS2). Large chemical libraries can be screened computationally to identify promising therapeutic candidates for viral entrance and replication. The pace of target identification and structural determination continues to accelerate, driven by technological advancements. These developments in structure-based drug design (SBDD) are crucial for identifying lead compounds targeting key receptors, not only for the current pandemic but also to prepare for future health emergencies.

Overall, in silico study has helped in the discovery of possible therapeutic candidates, the creation of small molecule inhibitors, and the forecasting of the behavior of the virus and its interactions with human cells, in addition to suggesting possible treatment candidates and taking mutations into consideration [129,130,131]. Ultimately, collaboration among diverse research groups, universities, pharmaceutical companies, and regulatory agencies is paramount. This collaborative effort accelerates drug discovery and ensures that effective COVID-19 treatments swiftly reach the public. The urgency of such collaboration is underscored by the ongoing fight against SARS-CoV-2 and the potential health challenges of the future.

## Figures and Tables

**Figure 1 ijms-24-15518-f001:**
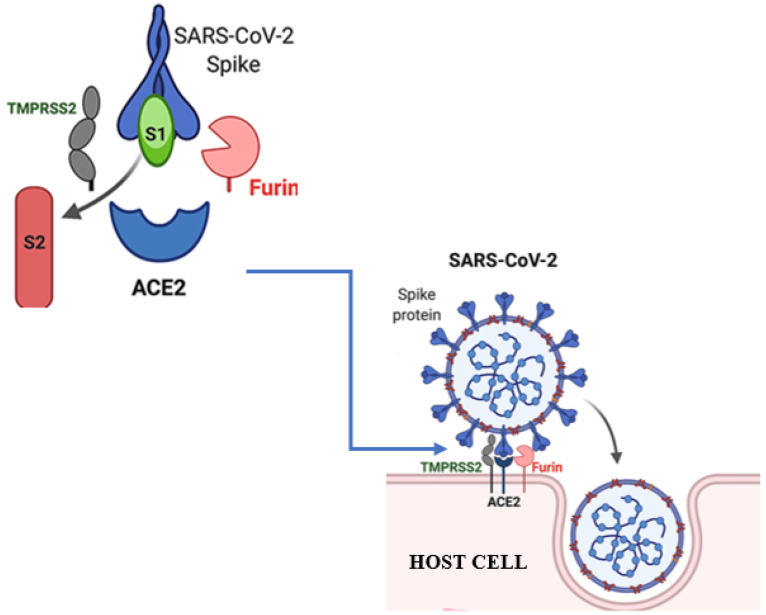
The cleaving of SARS-CoV-2 receptors, as adapted from a source: The viral spike protein of the new coronavirus SARS-CoV-2 uses the same cellular receptor (ACE2) as SARS-CoV and requires the cellular protease TMPRSS2 for its activation [25]. The image was created with BioRender.

**Figure 2 ijms-24-15518-f002:**
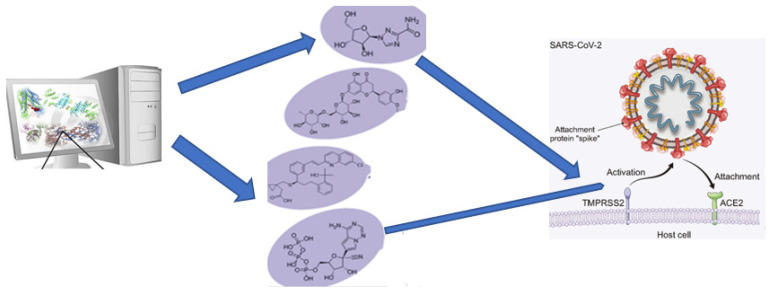
Targeting SARS-CoV-2 receptor binding domain, as adapted from a source: SARS-CoV-2 causes a reduction in the expression of the ACE2 receptor, without affecting ACE, by interacting with the ACE2 receptor through the spike protein. This interaction facilitates the virus’s entry into cells and its replication, leading to severe lung damage. Potential therapeutic strategies include the development of a vaccine based on the SARS-CoV-2 spike protein, the use of a TMPRSS2 inhibitor to prevent spike protein activation, the blockade of the surface ACE2 receptor using anti-ACE2 antibodies or peptides, and the use of a soluble ACE2 form. The soluble ACE2 can compete with SARS-CoV-2 for binding, reducing viral entry into cells, and safeguarding lung tissue from injury due to its unique enzymatic function [32].

**Figure 3 ijms-24-15518-f003:**
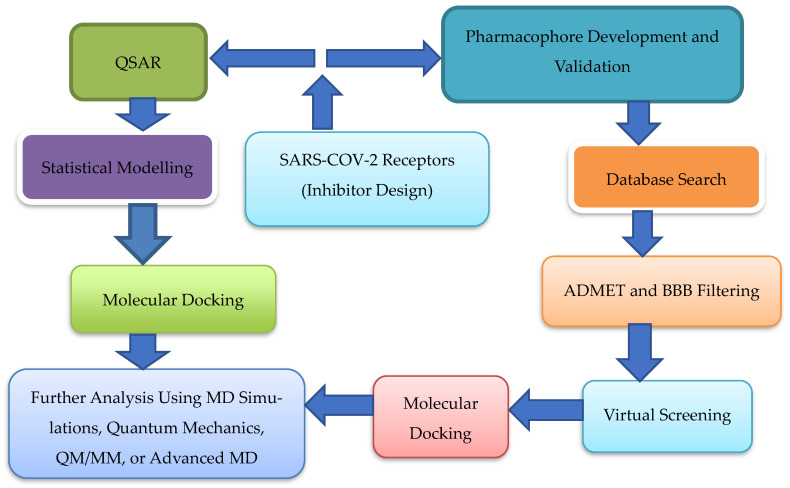
Redrawn schematic illustration of prevalent computational methods utilized for inhibition design of SARS-CoV-2 receptors (such as ACE2, TMPRSS2, S protein, and main protease) as adapted from source [47].

## Data Availability

No new data were created or analyzed in this study. Data sharing is not applicable to this article.

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
