# Peer review of "Assessing the Potential Contribution of In Silico Studies in Discovering Drug Candidates That Interact with Various SARS-CoV-2 Receptors"

_ijms, 2023, doi:10.3390/ijms242115518_

Round 1

Reviewer 1 Report

In attention of the manuscript authors,

The ”ijms-2566771-peer-review-v1” manuscript aim is to evaluate the contribution and importance of in silico studies to discover/develop novel candidates that interact with SARS-CoV-2 receptors. In this regard, the authors surveyed the literature between 2019 and 2023 and systematized information to provide an overview of SARS-CoV-2 research progress. The results provided by the manuscript could be a real gain for researchers interested to study SARS-CoV-2 and identify promising candidates to fight against the virus.

In this context, the potential impact of the manuscript results in the research world, and with all due respect to the author’s extensive work to find and summarized the information, the manuscript may be considered for publication in the IJMS journal following major revision.

Even if the manuscript provides several interesting observations, there are still critical issues that need to be addressed, such as:

1. The introduction section could be improved in terms of English. There are repetitive terms and descriptions that complicate the text and reduce its clarity, relevance, and importance. Moreover, the main purpose of the paper is not clear and presented in an advantageous way.

2. There are 3 sections that present almost the same information. Furthermore, in sections, 2.3 2.4, and 2.5 are observed identical sentences such as:

Section 2.3: lines 155-159 with Section 2.4: lines 177-181

Section 2.4: lines 166-168, 169-172 with Section 2.5: lines 192-195, 196-199

I strongly recommend to analyze and combine the information of sections 2.1 - 2.5 into a reduced number of sections and keeping only the relevant information. Please avoid the presence of identical sentences and implicitly the redundant information.

3.  Almost the same information are presented in the sections 3, 3.1, and 3.2. Please rewrite these sections and keep the relevant information. Short sections which present significant information for the manuscript's purpose are requested.

4.  Why was the information in section 4 not associated with that in the 4.1 section? Section 4.1, also, includes molecular dynamic discussions.

5.  The above-mentioned observation are spread over the entire manuscript text. I strongly recommend to systematize the information in a more clear and easy to follow version.

6. The conclusion must be rewritten in a more convincing way. The general impression is that the authors merely selected articles and presented the information without any input of their own.

The manuscript's English could be improved.

Author Response

REVIEWER 1

General comment: The manuscript's aim is to evaluate the contribution and importance of in silico studies to discover/develop novel candidates that interact with SARS-CoV-2 receptors. In this regard, the authors surveyed the literature between 2019 and 2023 and systematized information to provide an overview of SARS-CoV-2 research progress. The results provided by the manuscript could be a real gain for researchers interested in studying SARS-CoV-2 and identify promising candidates to fight against the virus.

In this context, the potential impact of the manuscript results in the research world, and with all due respect to the author’s extensive work to find and summarized the information, the manuscript may be considered for publication in the IJMS journal following major revision.

Even if the manuscript provides several interesting observations, there are still critical issues that need to be addressed, such as:

Comment 1: The introduction section could be improved in terms of English. There are repetitive terms and descriptions that complicate the text and reduce its clarity, relevance, and importance. Moreover, the main purpose of the paper is not clear and presented in an advantageous way.

Response 1: Thank you for the constructive comment. Repetitive terms and descriptions have been removed. The introduction section has been improved to read better now. See line 39-108 (page 1&2). The main purpose was improved as well on lines 88-93 (page 2).

Comment 2: There are 3 sections that present almost the same information. Furthermore, in sections, 2.3 2.4, and 2.5 are observed identical sentences such as:

Section 2.3: lines 155-159 with Section 2.4: lines 177-181

Section 2.4: lines 166-168, 169-172 with Section 2.5: lines 192-195, 196-199

I strongly recommend to analyze and combine the information of sections 2.1 - 2.5 into a reduced number of sections and keeping only the relevant information. Please avoid the presence of identical sentences and implicitly the redundant information.

Response 2: Thank you for the constructive comment. The redundant parts of the section have been removed and subsections all combined in one section for the paper to read better. Refer to lines 109-188 (pages 3-5).

Comment 3:  Almost the same information is presented in the sections 3, 3.1, and 3.2. Please rewrite these sections and keep the relevant information. Short sections which present significant information for the manuscript's purpose are requested.

Response 3: Thank you for the constructive comment. The redundant part of the section has been removed and subsections all combined in one section for the paper to read better. Refer to lines 189-266 (Pages 5-7)

Comment 4:  Why was the information in section 4 not associated with that in the 4.1 section? Section 4.1, also, includes molecular dynamic discussions.

Response 4: Thank you for the constructive comment. The section has been improved and divided into subsection one which focuses on the main viral host cell entry receptors (see lines 268-303, page 6), and the other subsection focuses on therapeutic interventions used in the identification of drug candidates. See line 305-342 (page7-10)  

Comment 5:  The above-mentioned observation is spread over the entire manuscript text. I strongly recommend to systematize the information in a clearer and easier-to-follow version.

Response 5: Thank you for your suggestion. The observation has been amended and the text has been adjusted throughout the manuscript for it to read better. Repetitions have been removed.

Comment 6: The conclusion must be rewritten in a more convincing way. The general impression is that the authors merely selected articles and presented the information without any input of their own.

Response 6: Thank you for the constructive comment. The conclusion has been improved whereby authors have added current work by their study group. The section reads better now. See lines 376-424 (pages 11-12) in the updated version of the manuscript.

Reviewer 2 Report

Reviewer comments and suggestions

The authors in this study highlighted the use of in silico studies in the drug discoveries against SARS-Cov-2, which offers several advantages, including the ability to screen a large number of drug candidates in a relatively short amount of time, thereby reducing the time and cost involved in traditional drug discovery methods. Additionally, they also reported using in silico studies, they can allow for the prediction of the binding affinity of drug candidates to target receptors, providing insight into their potential efficacy. 

Overall, the manuscript was well written. However, a few concerns/comments needed to be explained/modified. 

  1. Line 34-36 Here the authors can add today data of infections and death caused by COVID.
  2. Comments for line 52 The authors could add more on this topic and get results from previous studies adding their reference as well.
  3. Lines 54-55 Already discuss it so no need to add the same points
  4. Line 62-63 How many papers they included that need to be mentioned. A Bioinformatics figure needed to be here.
  5. Line 68-69 Same points were repeated.
  6. Line 102-103, please explain it well with the help of cited references
  7. 2.3 and 2.4 sections should be merged and comprehensively discussed. Seems to provide a similar meaning.
  8. Line 174-175 Here the authors could maximize the different types of receptors.
  9. I think Table 1 not relates to COVID 19 and I urge the authors to think again to propose what kind of tables and figures needed to be in the manuscript and new analyses are warranted in this study. 
  10. The manuscript lacks new information and is also not extensively investigated to reach the standard of IJMS. This was a simple review and the authors needs to add more information and content so that it could judge by the reviewers and readers.
  11. Line 465-468 There should be some novelty from this study so that the clinicians or health care workers could use the insilico data.
  12. All references should be modified based on MDPI journal guidelines.

Author Response

REVIEWER 2

General comment: The authors in this study highlighted the use of in silico studies in the drug discoveries against SARS-Cov-2, which offers several advantages, including the ability to screen a large number of drug candidates in a relatively short amount of time, thereby reducing the time and cost involved in traditional drug discovery methods. Additionally, they also reported using in silico studies, they can allow for the prediction of the binding affinity of drug candidates to target receptors, providing insight into their potential efficacy.

Overall, the manuscript was well written. However, a few concerns/comments needed to be explained/modified.

Comment 1: Line 34-36 Here the authors can add today data of infections and death caused by COVID.

Response to comment 1: Thank you for the comment. The data of infections has been added at the latest updates upon resubmission of the paper. See line 39-42 in the updated version of the manuscript (page 1).

Comment 2: Comments for line 52 The authors could add more on this topic and get results from previous studies adding their reference as well.

Response to comment 2: We thank you for your suggestion. The section has been improved. See lines 60-71 (page 2)

Comment 3: Lines 54-55 Already discuss it so no need to add the same points

Response to comment 3: We appreciate your observation. The redundant section has been removed.

Comment 4: Line 62-63 How many papers they included that need to be mentioned. A Bioinformatics figure needed to be here.

Response to comment 4: Thank you for the constructive comment. We have inserted papers as well as our own publications and ongoing work on the topic. See lines 68-71 (page 2) and lines 72-87 (page 2)

Comment 5: Line 68-69 Same points were repeated.

Response to comment 5: Thank you for pointing that out. Redundant part has been removed from the text.

Comment 6: Lines 102-103, please explain it well with the help of cited references

Response to comment 6: We appreciate your suggestion. We have included cited references, see lines 94-100 (pages 2-3)

Comment 7: 2.3 and 2.4 sections should be merged and comprehensively discussed. Seems to provide a similar meaning.

Response to comment 7: Thank you for the constructive comment. The section has been merged into one. See lines 109-188 (Pages 3-5)

Comment 8: Lines 174-175 Here the authors could maximize the different types of receptors.

Response to comment 8: We appreciate your suggestion. Authors have added a few literatures on the receptors. See lines 153-188 (pages 4-5)

Comment 9: I think Table 1 not relates to COVID 19 and I urge the authors to think again to propose what kind of tables and figures needed to be in the manuscript and new analyses are warranted in this study.

Response to comment 9: Thank you for the constructive comment. The table has been removed and merged with table 2. The manuscript has been amended fully based on comments. It reads better now.

Comment 10: The manuscript lacks new information and is also not extensively investigated to reach the standard of IJMS. This was a simple review and the authors needs to add more information and content so that it could judge by the reviewers and readers.

Response to comment 10: We appreciate your feedback. The manuscript has been improved in its entirety from introduction to conclusion following the set of comments raised. It reads better now.

Comment 11: Lines 465-468 There should be some novelty from this study so that the clinicians or health care workers could use the in-silico data.

Response to comment 11: Thank you for the comment, the novelty has been added in the section. See lines 377-419, pages 11-12.

Comment 12: All references should be modified based on MDPI journal guidelines.

Response to comment 12: Thank you for the constructive comment. comment. Authors have revised all references in accordance with MDPI journal guidelines.

Round 2

Reviewer 1 Report

In attention of the journal Authors,

The authors responded satisfactorily to all the referee’s requirements and made all the changes addressed in the manuscript. The manuscript has been substantially improved in both chemical content and English. The results analysis, interpretation, and presentation employed to achieve their goal recommend this manuscript presenting scientific soundness and interest for the readers focused on triazoles and their use as possible agents to combat SARS-CoV-2.

In this context, I agree that the manuscript should be accepted for publication in the IJMS journal, in the current version.

Author Response

The authors wish to thank the reviewer for his constructive comments that have helped to reshape the review to read better. 

We thank the reviewer for his encouraging comment.

Reviewer 2 Report

Please check the figure 2 (written images need to fully seen) and table 2 (references should be in number)

I saw that authors put complete references in the text of the manuscript, it should be in numerical. What the red color indicate?

Author Response

The authors wish to thank the reviewer for his constructive comments that helped us to improve the review. it reads better now.

Comment 1: Please check figure 2 (written images need to be fully seen) and table 2 (references should be in number)

Response to comment 1: Thank you for the constructive comment. The image resolution has been improved. The table has been fully referenced with numbers.

Comment 2: I saw that authors put complete references in the text of the manuscript, it should be in numerical. What does the red color indicate?

Response to comment: Thank you for the comment. The authors did not see this part of the text. But we have checked the work and adjusted it to look better.